# Swept Source-Optical Coherence Tomography-Guided Facedown Posturing to Minimize Treatment Burden and Maximize Outcome after Macular Hole Surgery

**DOI:** 10.3390/jcm12165282

**Published:** 2023-08-14

**Authors:** Mariko Sato, Takeshi Iwase

**Affiliations:** Department of Ophthalmology, Akita University Graduate School of Medicine, Akita 010-8543, Japan; mariko.s@med.akita-u.ac.jp

**Keywords:** facedown positioning, gas-filled eye, macular hole, pars plana vitrectomy, spectral domain optical coherence tomography, swept source optical coherence tomography

## Abstract

We evaluated the closure of full-thickness macular holes (MHs) the day after surgery in minimizing the burden and maximizing patient outcomes. Herein, 25-gauge pars plana vitrectomy, internal limiting membrane peeling, and fluid–gas (20% sulfur hexafluoride) were performed for the treatment. Patients were instructed to remain in the facedown position until the confirmation of MH closure, and the position was discontinued in cases where the closure was confirmed. In total, 43 eyes of 43 patients, whose average age was 69.7 ± 8.6 years, were enrolled in this study. We used swept source (SS)-optical coherence tomography (OCT) for the confirmation of MH closure for gas-filled eyes and used spectral domain (SD)-OCT for the reconfirmation of MH closure after the gas volume was reduced to less than half of the vitreous cavity. MH closure was confirmed in 40 eyes (93%, the closure group) on the next day after surgery. The time from surgery to SS-OCT imaging was 24.7 h. Although facedown positioning was terminated in cases where MH closure was confirmed, there were no cases in which the MH was re-opened afterward. The basal and minimum MH size was significantly larger in the non-closure group than that in the closure group (*p* = 0.027, *p* = 0.043, respectively). Therefore, checking with SS-OCT the day after surgery and terminating facedown positioning in cases where MH closure was confirmed would be a useful method, removing a great burden for the elderly without sacrificing the MH closure rate.

## 1. Introduction

Kelly and Wendel first described the benefit of pars plana vitrectomy (PPV) and tamponade for a full-thickness macular hole (MH) in 1991 [1], and the results showed that the anatomical closure rate of a MH after PPV with 1 week of maintaining facedown posturing was 58%. Surgeons have attempted to further enhance patient outcomes and closure rates by implementing and attempting various modifications to the technique for repair [2,3,4,5,6], and this has increased the closure rate, resulting in a high success rate [7,8]. A 2015 Cochrane Database of Systemic Reviews concluded that PPV is effective in improving visual acuity and in achieving MH closure versus the observation [9].

Optical coherence tomography (OCT) is akin to noninvasive tissue “biopsy”—it provides in vivo cross-sectional views (tomography) of internal tissue structures similar to tissue sections under a microscope [10]. OCT is vital for ophthalmologic clinical and surgical decision-making, in particular for macular pathologies. It complements clinical examination in diagnosing vitreoretinal interface pathologies, including macular holes [11]. OCT allows the clinician to detect initial stages of MH, follow its progression, and intervene early in case of progression to full-thickness MHs. It can unequivocally detect the presence of a MH as well as changes in the surrounding retina, distinguishing it from lamellar holes and cystic lesions of the macula. Additionally, the status of the vitreomacular interface can be evaluated. Various MH factors enable the surgeon to discuss the prognosis with patients to provide a more realistic expectation. Novel surgical modifications have been attempted for large macular holes diagnosed on OCT, with improved postoperative results. Postoperative evaluation with OCT helps to elucidate the structural and functional changes associated with different surgical techniques. It helps us understand the mechanisms of postoperative improvement observed along with changes in the retinal architecture. However, the inability to visualize the macula in gas-filled eyes during early postoperative periods can be one of the most important obstacles for the evaluation of the sealing process in eyes with MH. If OCT permits early postoperative evaluation of the macula in gas-filled eyes, then it may provide useful clinical information that allows the retinal surgeon to make a clinical decision about when the patient can release the facedown position. Several reports have used spectral domain OCT (SD-OCT) to determine whether the MH was closed in gas-filled eyes [12,13,14]. SD-OCT uses light sources of wavelengths near 800 nm, whereas swept source OCT (SS-OCT) uses those longer than 1000 nm. The difference leads SS-OCT to perform significantly better than SD-OCT to visualize the macula in gas-filled eyes [15].

During the postoperative period, a key variable that can be modified is facedown posturing. When Kelly and Wendel first introduced PPV for MH treatment, they required that the patients stay facedown posturing for one week [1]. Thus, facedown posturing has become a standard of MH treatment. The rationale for facedown posturing is to ensure the isolation of full-thickness MH from the intraocular fluid, which would permit the absorption of subfoveal fluid and lead to the apposition of the edges and closure of the full-thickness MH [16]. The evidence suggests that the first 24 h is likely the most critical time period for isolating the full-thickness MH from the intraocular fluid [17]. A meta-analysis suggested a significantly higher all-size full-thickness MH closure rate in the facedown posturing group compared with the non-facedown posturing group [18]. Keeping in line with this finding, most surgeons continue to recommend facedown posturing. A survey by the American Society of Retina Specialists in 2020 demonstrated that >85% of US retinal surgeons still recommend >2 days of facedown posturing, and >95% of US retinal surgeons still incorporate facedown posturing in clinical practice.

However, facedown posturing can be a great burden for the elderly, with physical, emotional, and medical risks. Unfortunately, idiopathic MH typically occurs in older people who are least able to maintain such a position owing to health issues [19]. Over 50% of patients have described facedown posturing as difficult or very difficult, and strict patient compliance with facedown posturing instructions is not often achieved [20]. In addition, rare but serious systemic complications, such as thrombophlebitis, pulmonary embolism, or ulnar nerve palsies, have been reported after facedown posturing [21]. Therefore, it is desirable to decrease the amount of time spent in this position.

Modifications of facedown posturing have been widely debated, and there have been proposals to shorten the duration or eliminate facedown posturing completely [22,23,24,25,26]. Tornambe et al. first stated that a facedown posture was not necessary, and other studies have suggested a shortening of the posturing period [22]. A multicenter randomized controlled study showed that the MH closure rate without facedown posturing was 90% for small idiopathic MHs [21]. However, there are still concerns that shortening posturing or non-posturing may lead to failure in some eyes [27]. Currently, there is no consensus among clinicians regarding the ideal positioning requirements after full-thickness MH surgery.

To solve those problems, facedown posturing should be released after confirming that the MH is closed. Shah et al. proposed the concept of “OCT-guided facedown positioning” [12]. This method was very effective in reducing positioning time without sacrificing the MH closure rate since the hole was less likely to reopen after early postoperative filling. In that study, however, SD-OCT was used, and the closure of the MH should be more accurately confirmed using SS-OCT. Since the first 24 h is likely the most critical time period for isolating the full-thickness MH from the intraocular fluid [17], more cases may be released from the facedown posturing earlier if the closure of the MH is confirmed using SS-OCT the day after surgery. This method would minimize the burden of the facedown positioning and maximize its effectiveness.

The purpose of this study was to evaluate the closure of MHs by SS-OCT the day after surgery to minimize the burden of facedown positioning and maximize its effectiveness. Facedown positioning was terminated in cases where MH closure was confirmed using SS-OCT, and facedown positioning was continued in cases where closure was not confirmed. After the gas volume was reduced to less than half of the vitreous cavity, the closure of the MH was reconfirmed using SD-OCT.

## 2. Patients and Methods

This was a retrospective cross-sectional, single-center study. The Ethics Committee of Akita University Hospital (Akita, Japan) approved the procedures, and the procedures conformed to the tenets of the Declaration of Helsinki. Informed consent was obtained from all participants after explaining the nature and possible complications of the study.

### 2.1. Subjects and Testing Protocol

We reviewed patients who had been diagnosed with MH and were surgically treated between January 2021 and August 2022 at the Department of Ophthalmology of Akita University Hospital. All patients underwent a comprehensive ophthalmic examination, including measurements of best-corrected visual acuity (BCVA), intraocular pressure (IOP), axial length, slit-lamp examination, and fundus examination. Snellen VA values were converted into the logarithm of the minimum angle of resolution (LogMAR) units to generate a linear scale of VA.

High myopia [28], preexisting macular conditions [29], and other factors may affect the anatomical success rate of MH closure. Exclusion criteria were a high myopia (axial length ≥ 27 mm), preexisting macular conditions (e.g., epiretinal membrane (ERM), macular degeneration, vascular occlusive diseases, or diabetic retinopathy), secondary MH, history of vitrectomy, and inability to maintain posturing. Patients were also excluded if their SS-OCT measurements showed poor scan quality [30] or artifacts (defocus, blink lines, or motion artifacts).

### 2.2. Surgical Techniques

Standard three-port pars plana vitrectomy (PPV) was performed by a single surgeon (T.I.) with 25-gauge (G) instruments. To begin the PPV procedure, a trocar was inserted at an approximate angle of 30° to the limbus. Once the trocar was past the trocar sleeve, the angle was changed to be perpendicular to the retinal surface. After creating the three ports, PPV was performed using the Alcon Constellation system (Alcon Laboratories, Inc., Fort Worth, TX, USA). After completion of the core vitrectomy, a posterior vitreous detachment was created if it was not present. Then, the ILM was simply peeled from the retina with a diameter of about two optic discs using ILM-peeling forceps (25G ILM forceps MAS25-25; HOYA surgical optics, Gamagori, Aichi, Japan) assisted by triamcinolone acetonide; the inverted ILM flaps technique was not used in all cases. Fluid–air exchange was performed; then, 20% sulfur hexafluoride (SF_6_) was injected into the vitreous upon completion of the PPV. After the IOP was adjusted to normal levels, the cannulas were withdrawn, and the sclera was pressed and massaged with an indenter to close the wound. The patients were instructed to maintain a facedown position after surgery. Facedown positioning was terminated in cases where MH closure was confirmed using SS-OCT, and facedown positioning was continued in cases where the closure was not confirmed.

Cataract surgery was performed on all phakic eyes, and a foldable acrylic IOL was implanted into the capsular bag.

### 2.3. SS-OCT Imaging

A PLEX Elite^®^ (Carl Zeiss Meditec, Dublin, CA, USA) is a SS-OCT instrument that uses a swept laser source with a central wavelength from 1040 to 1060 nm (980–1120 nm full bandwidth) and operates at 100,000 A-scans per second. The SS-OCT devices provide line scans. In this study, high-quality 6 mm horizontal line scans were obtained through the fovea and used for analyses. All the OCT scans were performed twice to minimize the possibility of accidental poor performance during OCT examination, and the higher-quality images were used for analyses. The closure of MH was confirmed using the SS-OCT from the next day after surgery. The patients were divided into two groups, a MH closure group and a non-closure group, using SS-OCT examination, depending on whether the MH was closed on the next day after surgery. The closure of MH was confirmed daily thereafter using SS-OCT until the gas volume was reduced to less than half of the vitreous cavity.

### 2.4. SD-OCT Imaging

A Spectralis^®^ SD-OCT (Heidelberg Engineering, Heidelberg, Germany) was used to obtain all SD-OCT images before surgery and after the gas volume was reduced to less than half of the vitreous cavity. The MHs were graded according to the Gass classification [31]. The images of the retina obtained by SD-OCT B scans through the center of the MH were analyzed. The basal MH size and the minimum MH size were measured in the vertical and horizontal OCT images (Figure 1). The average of two measurements was used for the statistical analyses. The MH closure was reconfirmed using SD-OCT after the gas volume was reduced to less than half of the vitreous cavity. Subsequently, SD-OCT was taken at 1 month and 6 months postoperatively to confirm the MH closure.

### 2.5. Statistical Analyses

All statistical analyses were conducted using IBM SPSS Statistics for Windows, Version 26.0 (IBM Corp., Armonk, NY, USA). Data are presented as the mean ± standard deviation. The paired *t*-test was used to compare the visual acuity before and after surgery. The unequal n *t*-test was used to compare the MH size data between the closure and non-closure group. A *p* value of <0.05 was considered significant.

## 3. Results

### 3.1. Clinical Characteristics of the Subjects

In total, 54 eyes of 54 Japanese patients with a MH underwent vitrectomy with ILM peeling between January 2021 and August 2022. Of these, eleven eyes were excluded: six for ERM, one for secondary MH from a rupture of retinal arterial macroaneurysm, one for branch retinal vein occlusion, one for retinoschisis, and two for poor scan quality on the next day after surgery. In the end, 43 eyes of 43 patients (mean age, 69.7 ± 8.6 years) were studied. The demographic and baseline characteristics are summarized in Table 1.

The number of eyes with MH stage 1 was four, stage 2 was twelve, stage 3 was fifteen, and stage 4 was twelve. The mean axial length was 23.92 ± 1.43 mm. The basal MH size and the minimum MH size were 677.7 ± 318.2 μm and 280.7 ± 165.2 μm, respectively. The preoperative BCVA was 0.69 ± 0.33 logMAR units, and it was significantly improved to 0.24 ± 0.40 logMAR units after surgery (*p* < 0.001). Cataract surgery was performed on all 37 phakic eyes.

### 3.2. Macular Hole Closure Rate after Surgery

MH closure was confirmed in 40 eyes (93%) on the next day after surgery (Figure 2). The time from surgery to SS-OCT imaging was 24.7 ± 4.6 h. In the remaining three eyes, MH closure could not be confirmed in the first few postoperative days, though the closure of MH was confirmed daily thereafter using SS-OCT. As a result, in the three cases in which MH closure could not be confirmed the day after surgery, the closure was confirmed by SD-OCT after the gas volume was reduced to less than half of the vitreous cavity (Figure 3). The MH was closed in 41 eyes (96%) on postoperative day 7 and in 43 eyes (100%) on postoperative day 10. The MH closure was reconfirmed using SD-OCT after the gas volume was reduced to less than half of the vitreous cavity in eyes in which MH closure was confirmed the day after surgery. All eyes were followed up for 6 months after surgery, and it was confirmed that the MH was closed. Thus, all eyes with MH were successfully closed after the initial surgery in this study.

### 3.3. Comparison of the MH Size between the Closure Group and the Non-Closure Group on the Next Day after Surgery

The basal MH size was 1067.5 ± 155.9 μm in the non-closure group and 648.4 ± 315.5 μm in the closure group, and it was significantly larger in the non-closure group than that in the closure group (*p* = 0.027) (Figure 4). The minimum MH size was 467.3 ± 240.1 μm in the non-closure group and 267.2 ± 156.9 μm in the closure group, and it was significantly larger in the non-closure group than that in the closure group (*p* = 0.043).

## 4. Discussion

MH closure was confirmed in 40 eyes (93%) using SS-OCT in MHs on the next day after surgery. The time from surgery to SS-OCT imaging was 24.7 h. Facedown positioning was terminated in cases where MH closure was confirmed, and there were no cases in which the MH was reopened afterward. Eyes with no closure of the MH on the next day tended to have larger preoperative MH basal and minimum MH size.

After Kelly and Wendel first described the treatment for full-thickness MHs [1], surgeons have attempted to further enhance patient outcomes and closure rates by implementing and attempting various modifications to the technique for repair [2,3,4,5,6]. With the modifications of the surgical technique, such as ILM peeling [2,3,4], inverted ILM flaps [5,6], autologous serum use [32], amniotic membrane [33], and retinal grafts [34], MH has become a treatable disease with a high anatomical success rate reaching more than 90% [7,8]. Despite these improvements, there is continued debate over whether postoperative posturing is necessary. Since the immobility arising from prolonged postoperative facedown positioning negatively impacts the quality of life of patients and causes occasional but severe adverse events [21,35], there has been an attempt to alleviate patient discomfort by shortening the duration of posturing [14,36,37,38]. Tornambe et al. first stated that facedown posturing was not necessary [22,23,24,25]. In addition, a multicenter randomized controlled study showed that the MH closure rate without facedown posturing was 90% for small idiopathic MHs [21]. Those results indicate that there are still concerns that shortening or non-posturing might lead to failure in some eyes.

Some investigators have utilized postoperative OCT analysis in gas-filled eyes to confirm the early closure of the MH and to individualize the period of posturing [12,14,26,38,39]. Shah et al. proposed the concept of “OCT-guided facedown positioning” [12]. This method was very effective in reducing positioning time without sacrificing the MH closure rate, since the hole was less likely to reopen after early postoperative filling. In that study, however, SD-OCT was used. The closure of MH was determined by SD-OCT in between 75 and 86% of cases on postoperative day 1 [12,40]. In our study, 93% of MH eyes were determined to be closed by SS-OCT on postoperative day 1. It has been reported that SS-OCT performs better than SD-OCT for visualization of the macula in gas-filled eyes [15]. Although we cannot directly compare our results with previous reports [12,40] because of the different backgrounds of the patients, the use of SS-OCT may be one of the reasons for the high number of cases in which closure of the MH at postoperative day 1 was confirmed in this study. In addition, SS-OCT may be the better choice for imaging used for OCT-guided facedown positioning in gas-filled eyes.

The MH size was significantly larger in the non-closure group than that in the closure group in our study. In general, the prognosis for closure is correlated with the size of the hole [41,42]. Recently, it has been reported that the MH size is significantly and positively correlated with the time of closure [14], meaning that the larger the MH size, the longer the time until closure. In the present study, there was also a trend that the MH did not close on postoperative day 1 in cases with larger MH sizes; in cases where the MH did not close on postoperative day 1, MH closure was confirmed at a maximum of 10 days. However, the MH closure was difficult to confirm when the gas volume was a little more than half of the vitreous cavity; although the MH closure was confirmed at 10 days, it may have actually occurred a little earlier. Since the inverted flap technique was not used in this study, it may have taken longer to close the MH [43]. In cases with larger MH sizes, preoperative explanation of the possibility of a prolonged period of facedown positioning and the use of the inverted flap technique should be considered.

After confirming closure of the MH by SS-OCT on the next day after surgery, the MH did not reopen, although facedown positioning was terminated after the confirmation. Reopening of the MH after surgery has been reported in between 2% and 10% of cases in the literature [44]. Ip et al. reported that reopening larger than 400 μm was seen in MHs [41]. In the present study, the MH size tended to be smaller in patients with MH closure on the day after surgery. As mentioned earlier, it has been reported that the MH size is significantly and positively correlated with the time of closure [14]. Therefore, once closure of the MH was confirmed the day after the surgery, it was unlikely that the MH would have reopened even if facedown positioning was terminated, since the MH was not large in many of these cases.

This study has some limitations. First, this was a retrospective study with a relatively small sample size. Second, although the MH closure was confirmed daily after surgery, the MH closure was difficult to confirm when the gas volume was a little more than half of the vitreous cavity. Perhaps it would be more accurate to check the time of MH closure by taking frequent SS-OCT images; e.g., every 6 h after surgery. However, this is a retrospective study, and it is not usual to take images frequently. Third, the inverted flap technique was not used for the treatment of MH in this study. It has been reported that the inverted flap technique requires a shorter time and achieves higher success rates in closing larger MHs. The present method might be more effective because the MH closure may be obtained more quickly and reliably using this technique. Further randomized controlled trial studies on a larger number of eyes with more frequent imaging to examine the closure time of the MH after surgery are required to confirm our method.

## 5. Conclusions

Closure was confirmed in 93% of eyes with MH using SS-OCT on the day after surgery, and facedown positioning was terminated in cases where MH closure was confirmed using SS-OCT. In conclusion, this present method is useful in that it shortens the period of facedown positioning without causing the recurrence of MH, which is a great burden for the elderly.

## Figures and Tables

**Figure 1 jcm-12-05282-f001:**
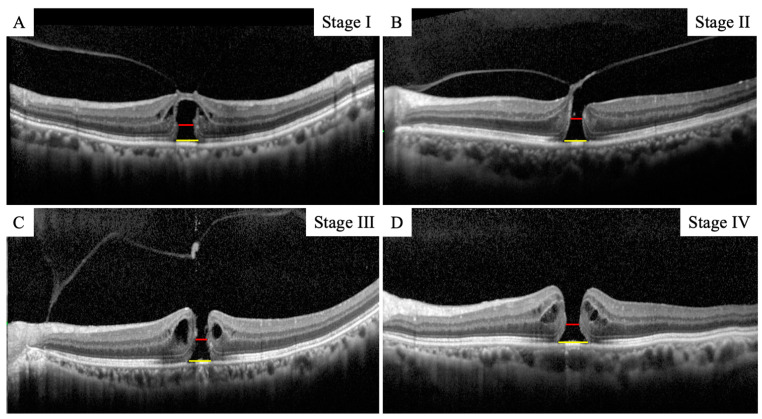
Representative image of measurement of the basal MH size and the minimum MH size at all stages (**A**–**D**). The red line indicates the basal MH size and the yellow line indicates the minimum MH size.

**Figure 2 jcm-12-05282-f002:**
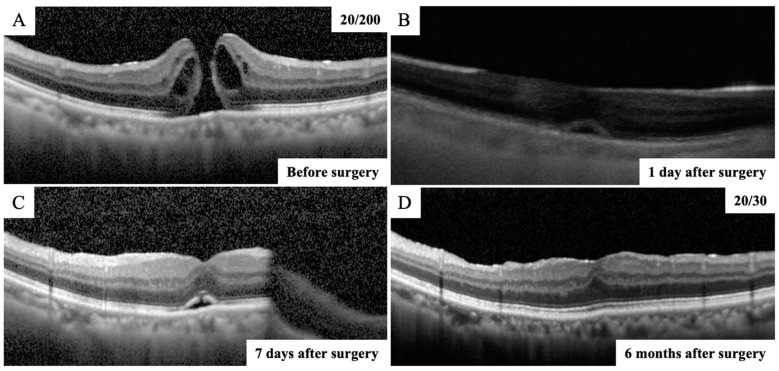
OCT images of vertical scans of the right eye of a 74-year-old man with a MH. (**A**) The preoperative SD-OCT image showing an MH and the right vision was 20/200. (**B**) SS-OCT image through complete intraocular gas endotamponade on postoperative day 1 showing that the MH was closed with a continuous outer retinal layer at the fovea. (**C**) After the gas volume was reduced to less than half of the vitreous cavity, the closure of the MH was reconfirmed using the SD-OCT on postoperative day 7. (**D**) Postoperative SD-OCT image demonstrated a closed MH with continuous ellipsoid zone band at the fovea and the right vision improved to 20/30 at 6 months after surgery.

**Figure 3 jcm-12-05282-f003:**
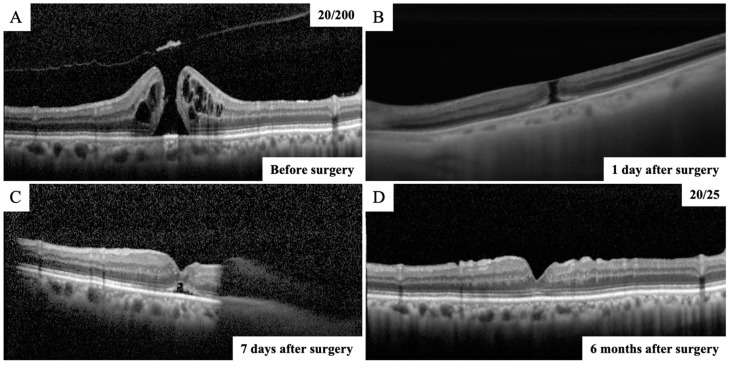
OCT images of vertical scans of the right eye of a 68-year-old man with a MH. (**A**) The preoperative SD-OCT image showing an MH and the right vision was 20/200. (**B**) SS-OCT image through complete intraocular gas endotamponade on postoperative day 1 showing that the MH was still open. (**C**) After the gas volume was reduced to less than half of the vitreous cavity, the SD-OCT image showed that MH was closed on postoperative day 7. (**D**) Postoperative SD-OCT image demonstrated a closed MH with continuous ellipsoid zone band at the fovea and the right vision improved to 20/25 at 6 months after surgery.

**Figure 4 jcm-12-05282-f004:**
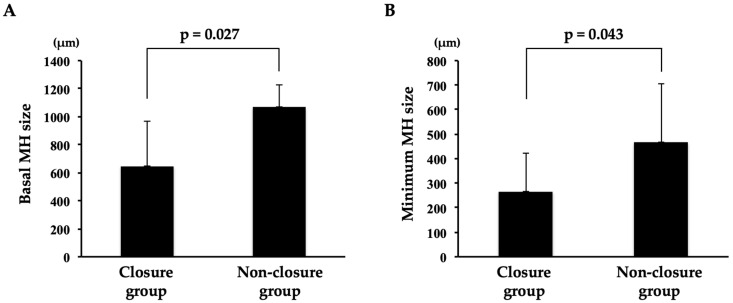
Comparison of basal and minimum MH size between the closure group and the non-closure group. (**A**) The basal MH size was significantly larger in the non-closure group than that in the closure group (*p* = 0.027). (**B**) The minimum MH size was significantly larger in the non-closure group than that in the closure group (*p* = 0.043).

**Table 1 jcm-12-05282-t001:** Clinical characteristic of the MH eyes.

Clinical Characteristic	
Age (year)	69.7 ± 8.6 (49–87)
Sex (male/female)	14/29
MH stage 1:2:3:4 (eyes)	4:12:15:12
Axial length (mm)	23.92 ± 1.43
Basal MH size (μm)	677.7 ± 318.2
Minimum MH size (μm)	280.7 ± 165.2
Preoperative BCVA (LogMAR)	0.69 ± 0.33
Preoperative BCVA (LogMAR)	0.24 ± 0.39
PPV + PEA + IOL/PPV (eyes)	37/6

## Data Availability

The data that support the findings of this study are available from the corresponding author upon reasonable request.

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
