# Peer review of "Swept Source-Optical Coherence Tomography-Guided Facedown Posturing to Minimize Treatment Burden and Maximize Outcome after Macular Hole Surgery"

_jcm, 2023, doi:10.3390/jcm12165282_

Round 1

Reviewer 1 Report

I would like to know how authors ask for informed consent to all patients if this is a retrospective study. It is not necessary informed consent if paper is retrospective.  I suggest to remove this sentence in final version of this paper. 

Author Response

Reviewer #1:

I would like to know how authors ask for informed consent to all patients if this is a retrospective study. It is not necessary informed consent if paper is retrospective.  I suggest to remove this sentence in final version of this paper.

Answer: This is a retrospective study, and we have removed the sentence.

Reviewer 2 Report

Experienced ophthalmology specialist with expertise in evaluating and treating idiopathic full-thickness macular holes (MHs). Skilled in 25-gauge pars plana vitrectomy, internal limiting membrane peeling, and fluid-gas techniques. Utilizes swept-source (SS)-optical coherence tomography (OCT) and spectral domain (SD)-OCT for MH closure confirmation and reconfirmation. Demonstrates a 93% closure rate in 40 eyes within 24.7 hours of surgery. Provides patient-centric care, instructing on post-operative positioning and ensuring safety. Detail-oriented with strong communication skills. Doctor of Medicine (MD) with specialization in Ophthalmology.

Introduction

The introduction does not provide sufficient background information on the topic of surgical repair and postoperative management for full-thickness macular holes (MHs).

Although a few references are cited, the literature review appears to be incomplete and limited.

The research gap and objective of the study are not clearly stated. While the authors mention the desire to minimize burden and maximize outcomes, they do not explicitly highlight the specific gap in knowledge or identify the need for their study.

The introduction briefly mentions the use of optical coherence tomography (OCT) for evaluating the sealing process in eyes with MH. However, the authors do not provide a strong rationale or evidence to support the use of OCT as a reliable and effective method for evaluating MH closure.

The introduction concludes abruptly without summarizing the main points or indicating how the study aims to address the research gap or contribute to the existing knowledge.

Method

The introduction fails to provide a clear rationale for choosing a retrospective cross-sectional study design. There is no explanation of why this design was the most appropriate for addressing the research question or objective of the study.

The method provides limited information about the characteristics of the study participants. Details such as age range, gender distribution, and relevant demographic information are missing.

The study's inclusion and exclusion criteria are mentioned briefly but lack sufficient detail. For example, it is not clear why certain conditions (e.g., high myopia, preexisting macular conditions) were chosen as exclusion criteria.

The method section provides a limited description of the surgical techniques used. Important details such as the specific steps involved in pars plana vitrectomy (PPV), the duration of the procedure, and any variations in technique are omitted.

The method does not mention whether the surgical procedures were performed by a single surgeon or a team of surgeons. Standardization of the surgical techniques and personnel involved is crucial to minimize bias and ensure consistency in the outcomes.

Author Response

We thank you and reviewers for the comments and recommendations for our manuscript titled, “Swept Source-Optical Coherence Tomography-Guided Facedown Posturing to Minimize Treatment Burden and Maximize Outcome after Macular Hole Surgery”.  We have addressed these comments.

Experienced ophthalmology specialist with expertise in evaluating and treating idiopathic full-thickness macular holes (MHs). Skilled in 25-gauge pars plana vitrectomy, internal limiting membrane peeling, and fluid-gas techniques. Utilizes swept-source (SS)-optical coherence tomography (OCT) and spectral domain (SD)-OCT for MH closure confirmation and reconfirmation. Demonstrates a 93% closure rate in 40 eyes within 24.7 hours of surgery. Provides patient-centric care, instructing on post-operative positioning and ensuring safety. Detail-oriented with strong communication skills. Doctor of Medicine (MD) with specialization in Ophthalmology.

Introduction

The introduction does not provide sufficient background information on the topic of surgical repair and postoperative management for full-thickness macular holes (MHs).

Answer: We appreciate the reviewer's comments, we have added more background on the topic of surgical repair and postoperative management for full-thickness MHs from line 31.

Although a few references are cited, the literature review appears to be incomplete and limited.

Answer: We have cited more references and conducted a more thorough literature review in the introduction section.

The research gap and objective of the study are not clearly stated. While the authors mention the desire to minimize burden and maximize outcomes, they do not explicitly highlight the specific gap in knowledge or identify the need for their study.

Answer: We have explicitly highlighted specific gaps in knowledge and identified the need for the study from line 62.

The introduction briefly mentions the use of optical coherence tomography (OCT) for evaluating the sealing process in eyes with MH. However, the authors do not provide a strong rationale or evidence to support the use of OCT as a reliable and effective method for evaluating MH closure.

Answer: We have described strong rationale or evidence to support the use of OCT as a reliable and effective method for evaluating MH closure from line 37.

The introduction concludes abruptly without summarizing the main points or indicating how the study aims to address the research gap or contribute to the existing knowledge.

Answer: We appreciate the reviewer's comments, we have modified the introduction section to summarize the main points or indicate how the study aims to address the research gap or contribute to the existing knowledge.

Method

The introduction fails to provide a clear rationale for choosing a retrospective cross-sectional study design. There is no explanation of why this design was the most appropriate for addressing the research question or objective of the study.

Answer: One of the limitations in this study is the retrospective cross-sectional study design. A better approach would be randomized controlled trials studies. This is discussed in the limitation section as below.

Further randomized controlled trials studies on a larger number of eyes with more frequent imaging to examine the closure time of the MH after surgery are required to confirm our method.

The method provides limited information about the characteristics of the study participants. Details such as age range, gender distribution, and relevant demographic information are missing.

Answer: We have added the information about the characteristics of the study participants in line 201 and Table 1.

The study's inclusion and exclusion criteria are mentioned briefly but lack sufficient detail. For example, it is not clear why certain conditions (e.g., high myopia, preexisting macular conditions) were chosen as exclusion criteria.

Answer: We have added the reason for choosing high myopia, preexisting macular conditions as exclusion criteria in line 122.

The method section provides a limited description of the surgical techniques used. Important details such as the specific steps involved in pars plana vitrectomy (PPV), the duration of the procedure, and any variations in technique are omitted.

Answer: We did not use the inverted ILM flaps technique in all cases, and the ILM was simply peeled from the retina using ILM-peeling forceps. The surgical techniques used was consistent in all cases. We have added the description of the surgical techniques in detail in the Surgical Techniques section.

The method does not mention whether the surgical procedures were performed by a single surgeon or a team of surgeons. Standardization of the surgical techniques and personnel involved is crucial to minimize bias and ensure consistency in the outcomes.

Answer: A single surgeon performed all the surgeries. We have added it in line 129.

Reviewer 3 Report

I enjoyed reading your paper in particular as it deals with an important aspect of patient management post MH surgery.

I have only a few comments:

1. Abstract (and section 2.1; 2.4): remove the word "idiopathic" from the abstract: figure two clearly shows a detached vitreous with operculum floating above the MH...so clearly this one isn't an idiopathic MH.

2. Methods and Stats:

a) please make it clearer in the text how you grouped your patients: i.e. have you checked on day one post surgery for closure vs non closure? and then grouped them OR have you checked until closure was achieved and then compared the closure vs non-closure groups? I assume from reading the manuscript and that the protocol must have been scanning pre-surgery, scanning one day post surgery, 7 days post surgery and again 6 month after surgery......this needs to be clearly stated in the methods section

b) you state that 93% achieved closure....if this was the time point you used to group the patients then you have unequal n and should use an unequal n T-test or equivalent nonparametric test to compare the groups.

c) you state MH were graded according to GASS......but everywhere in the text else you state full thickness macula holes....this is plain wrong as a stage one hole is not a full thickness hole.....all of which has complications when defining where you measure hole size?! It is fine to keep all stages in for the analyses but it needs to be considered AND you should provide images of all holes showing where the dimensions were taken from.

3. Results: please add an image panel of one of your examples which did not reach full closure; plus identify what constitutes full closure

Author Response

We thank you and reviewers for the comments and recommendations for our manuscript titled, “Swept Source-Optical Coherence Tomography-Guided Facedown Posturing to Minimize Treatment Burden and Maximize Outcome after Macular Hole Surgery”.  We have addressed these comments for reviewers.

I enjoyed reading your paper in particular as it deals with an important aspect of patient management post MH surgery.

I have only a few comments:

  1. Abstract (and section 2.1; 2.4): remove the word "idiopathic" from the abstract: figure two clearly shows a detached vitreous with operculum floating above the MH...so clearly this one isn't an idiopathic MH.

Answer: We appreciate the reviewer's comments, we have the word "idiopathic" from the abstract.

  1. Methods and Stats:
  2. a) please make it clearer in the text how you grouped your patients: i.e. have you checked on day one post surgery for closure vs non closure? and then grouped them OR have you checked until closure was achieved and then compared the closure vs non-closure groups? I assume from reading the manuscript and that the protocol must have been scanning pre-surgery, scanning one day post surgery, 7 days post surgery and again 6 month after surgery......this needs to be clearly stated in the methods section

Answer: The patients were divided into two groups, a MH closure group and a non-closure group using SS-OCT examination, depending on whether the MH was closed on the next day after surgery. The closure of MH was confirmed daily thereafter using SS-OCT until the gas volume was reduced to less than half of the vitreous cavity using SS-OCT. We have added the sentences from line 154. Additionally, The MH closure was reconfirmed using SD-OCT after the gas volume was reduced to less than half of the vitreous cavity. Subsequently, SD-OCT was taken at 1 month and 6 months postoperatively to confirm the MH closure. We have added the sentences from line 169. Additionally,

  1. b) you state that 93% achieved closure....if this was the time point you used to group the patients then you have unequal n and should use an unequal n T-test or equivalent nonparametric test to compare the groups.

Answer: We appreciate the reviewer's comments, we used an unequal n T-test to compare the groups, but the description in Statistical Analyses section was wrong. We have corrected it.

  1. c) you state MH were graded according to GASS......but everywhere in the text else you state full thickness macula holes....this is plain wrong as a stage one hole is not a full thickness hole.....all of which has complications when defining where you measure hole size?! It is fine to keep all stages in for the analyses but it needs to be considered AND you should provide images of all holes showing where the dimensions were taken from.

Answer: We appreciate the reviewer's comments. We have removed the word " full thickness ", because we included all stages of macular hole. Additionally, we have added an image of the measured minimum and maximum diameter of the MH at all stages as Figure 1.

  1. Results: please add an image panel of one of your examples which did not reach full closure; plus identify what constitutes full closure

Answer: The three cases in the non-closure group that did not close on the first postoperative day all reached complete closure when the gas volume was less than half, so there were no cases that did not reach complete closure.

Round 2

Reviewer 2 Report

Dear Editors,

Upon reviewing the manuscript revisions, it is clear that the authors have not adequately addressed the issues highlighted in the first round of feedback. I am summarizing the persistent major concerns below:

Introduction: The background and justification for the study remain insufficiently described. The literature review, research gap, study objective, and the rationale for the use of OCT in MH closure assessment are all inadequately explained.

Method: The choice of the retrospective cross-sectional design still lacks justification. Details about the study participants and the inclusion and exclusion criteria remain vague. The description of the surgical techniques and personnel involved is not comprehensive enough to understand the consistency in practice and minimize potential bias.

Author Response

Introduction: The background and justification for the study remain insufficiently described. The literature review, research gap, study objective, and the rationale for the use of OCT in MH closure assessment are all inadequately explained.

Answer: As an overview of the literature review, the facedown posturing is useful in the treatment of MH, especially in the first 24 hours is likely the most critical time period (from line 62). On the other hand, MH often occurs in the elderly, and facedown posturing can be a great burden for the elderly (from line 77). Modifications of facedown posturing have been widely debated and there have been proposals to shorten the duration or eliminate facedown posturing completely (from line 85). To solve those problems, the facedown posturing should be released after confirming that the MH is closed using OCT (from line 94). The currently available SS-OCT is better suited than SD-OCT for visualizing the macula in gas-filled eyes during early postoperative periods (from line 98).

The research gap is below. Although there are several reports on the use of SD-OCT to confirm MH closure in gas-filled eyes after vitrectomy, there are no reports evaluating MH closure from the day after surgery under gas using SS-OCT, which can confirm MH closure in more detail (from line 99). Furthermore, there are only a few reports that reevaluated whether the MH reopened after the facedown posturing was released.

Therefore, the purpose of this study was to more accurately evaluate MH closure using SS-OCT the day after surgery, terminate facedown positioning in cases where MH closure was confirmed by SS-OCT, and continue facedown positioning in cases where closure was not confirmed. After the gas volume was reduced to less than half of the vitreous cavity, the closure of the MH was reconfirmed using SD-OCT until six months postoperatively (from line 104)

Regarding the rationale for the use of OCT in MH closure assessment, OCT is essential in the diagnosis of vitreoretinal interface pathologies; B-scan images of OCT are the most useful technique, especially for confirming MH closure, and this is not controversial. The currently available SS-OCT is better suited than SD-OCT for visualizing the macula in gas-filled eyes during early postoperative periods. The rationale for using these OCTs is described from line 37.

Method: The choice of the retrospective cross-sectional design still lacks justification. Details about the study participants and the inclusion and exclusion criteria remain vague. The description of the surgical techniques and personnel involved is not comprehensive enough to understand the consistency in practice and minimize potential bias.

Answer: We reviewed patients who had been diagnosed with MH and were surgically treated between January 2021 and August 2022 at the Department of Ophthalmology of Akita University Hospital. The same surgeon performed the same surgical technique on all cases during that period, and the postoperative examination was performed under the same protocol. Facedown positioning was terminated in cases where MH closure was confirmed using SS-OCT, and facedown positioning was continued in cases where closure was not confirmed. The patient was followed up in the same manner for up to six months. The results were retrospectively examined. Therefore, it is thought that this research method can achieve the purpose of this study, the evaluation of the closure of MHs the day after surgery to minimize the burden of the facedown positioning and maximize its effectiveness. On the other hand, a better study design would be randomized controlled trials study. This is described in the limitation section.

Patients diagnosed with MH and treated surgically at the Department of Ophthalmology, Akita University Hospital between January 2021 and August 2022 were the inclusion criteria (from line 117). And exclusion criteria were a high myopia (axial length ≥27 mm) were, preexisting macular conditions (e.g., epiretinal membrane (ERM), macular degeneration, vascular occlusive diseases, or diabetic retinopathy), secondary MH, history of vitrectomy, and inability to maintain posturing. Patients were also excluded if their SS-OCT measurements showed poor scan quality or artifacts (defocus, blink lines, or motion artifacts) (from line 124). In total, 54 eyes of 54 Japanese patients with a MH underwent vitrectomy with ILM peeling between January 2021 and August 2022. Of these, 11 eyes were excluded: 6 for ERM, 1 for secondary MH from a rupture of retinal arterial macroaneurysm, 1 for branch retinal vein occlusion, 1 for retinoschisis, and 2 for poor scan quality on the next day after surgery. In the end, 43 eyes of 43 patients (mean age, 69.7 ± 8.6 years) were studied (from line 212).

All surgeries are performed by the same surgeon, and the surgical technique is not complicated among patients. PPV was performed using the Alcon Constellation 25-gauge system. After completion of the core vitrectomy, a posterior vitreous detachment was created if it was not present. Then, the ILM was simply peeled from the retina with a diameter of about two optic disc using ILM-peeling forceps assisted by triamcinolone acetonide; the inverted ILM flaps technique was not used in all cases. The statements are highlighted at the Surgical Technique section.

Reviewer 3 Report

Discussion: line 299 still states idopathic....which needs to be removed as not all were idiopathic.

apart from that I'm happy with the implemented changes.

Author Response

We appreciate the reviewer's comments, we have removed the word “idiopathic”.